# Course of Fecal Calprotectin after mRNA SARS-CoV-2 Vaccination in Patients with Inflammatory Bowel Diseases

**DOI:** 10.3390/vaccines10050759

**Published:** 2022-05-11

**Authors:** Jagoda Pokryszka, Angelika Wagner, Ursula Wiedermann, Selma Tobudic, Harald Herkner, Stefan Winkler, Sonja Brehovsky, Walter Reinisch, Gottfried Novacek

**Affiliations:** 1Department of Internal Medicine III, Division of Gastroenterology and Hepatology, Medical University of Vienna, 1090 Vienna, Austria; jagoda.pokryszka@meduniwien.ac.at (J.P.); sonja.brehovsky@akhwien.at (S.B.); gottfried.novacek@meduniwien.ac.at (G.N.); 2Institute of Specific Prophylaxis and Tropical Medicine, Medical University of Vienna, 1090 Vienna, Austria; angelika.wagner@meduniwien.ac.at (A.W.); ursula.wiedermann-schmidt@meduniwien.ac.at (U.W.); 3Department of Internal Medicine I, Division of Infectious Diseases and Tropical Medicine, Medical University of Vienna, 1090 Vienna, Austria; selma.tobudic@meduniwien.ac.at (S.T.); stefan.winkler@meduniwien.ac.at (S.W.); 4Department of Emergency Medicine, Medical University of Vienna, 1090 Vienna, Austria; harald.herkner@meduniwien.ac.at

**Keywords:** inflammatory bowel disease, SARS-CoV-2, fecal calprotectin, COVID-19 vaccine

## Abstract

Background: Two years into the pandemic, vaccination remains the most effective option to prevent coronavirus disease 2019 (COVID-19) caused by severe acute respiratory syndrome coronavirus 2 (SARS-CoV-2). Preliminary studies suggest vaccination efficacy in patients with inflammatory bowel diseases (IBD), but little is known about its impact on chronic intestinal inflammation. Here we assessed the mucosal inflammatory activity in patients with IBD before and after immunization with the mRNA-1273 (Moderna) vaccine by measurement of fecal calprotectin (fCP). Methods: In 42 patients with IBD, the baseline fCP levels obtained prior to the first vaccine were compared with the highest levels measured during and after two doses of vaccination. Patients’ sera were collected after the second dose to evaluate anti-SARS-CoV-2 antibodies’ titers. Results: We observed a significant fCP elevation in 31% of patients after any dose. Vedolizumab was identified as the only agent associated with an fCP increase (OR 12.4, 95% CI [1.6; 120.2], *p* = 0.0171). Gastrointestinal adverse events were reported in 9.5% of all subjects and in 75% of cases accompanied by an fCP increase. Anti-SARS-CoV-2 antibodies associated only weakly with the fCP increase after the first dose (*p* = 0.04). Conclusions: Our findings support possible collinearity in pathways of SARS-CoV-2 antigen expression and the pathogenesis of IBD.

## 1. Introduction

So far, for patients with inflammatory bowel diseases (IBD), an increased risk of infection with SARS-CoV-2 has not been demonstrated [1], though some medications, in particular corticosteroids, have been associated with an enhanced risk of severe COVID-19 outcome [2,3]. As immunomodulatory medication is known for putting the patients at an increased risk of serious infections [4], prevention of contagion plays a crucial role in their daily life. Despite the availability of SARS-CoV-2 vaccines and early evidence of their efficacy in patients with IBD, concerns about the safety related to immune-mediated disease enhancement need to be scientifically addressed [5,6]. Adverse events—including gastrointestinal symptoms—were reported from the clinical trials with mRNA vaccines [7,8] as well as in a series of IBD patients [9]. Increased stool frequency, abdominal pain, and rectal bleeding are the main symptoms of patients with IBD [10], but particularly the former two are less specific and could have various causes. Therefore, it would be relevant to know whether vaccination-induced gastrointestinal symptoms in patients with IBD are derived from an increase in intestinal inflammation and to shed light on their further course. Corresponding insights might inform the decision on the timing of vaccinations relative to the intestinal inflammatory burden.

During a routine follow-up in a patient with ulcerative colitis, we observed an increase in fecal calprotectin (fCP), the most sensitive and widely used biomarker for monitoring intestinal inflammation in patients with IBD, immediately after vaccination with the mRNA-1273 SARS-CoV-2 vaccine. Over the recent years, fCP has supplanted ileocolonoscopy, which, as an invasive procedure, is inadequate for rapid and repetitive monitoring in patients with IBD. We initiated the present study on patients with IBD, as a sub-study of a larger trial on the safety and immunogenicity of mRNA-1273 SARS-CoV-2 vaccine in immunocompromised patients, to explore changes in the intestinal inflammatory activity as measured by fCP during the course of the two-dose initiation SARS-CoV-2 vaccination.

## 2. Materials and Methods

Patients with IBD treated at the Medical University of Vienna and participated in a study on the safety and immunogenicity of the mRNA-1273 vaccine in immunocompromised patients were enrolled in our sub-study. Due to the exploratory character of the study, a formal sample size estimate was not performed. They received two doses of the mRNA-based SARS-CoV-2 vaccine (mRNA-1273, Moderna (Cambridge, MA, USA)) at an interval of 28 days. Patient and disease characteristics were obtained from electronic health records. Fecal calprotectin (fCP) levels were measured prior to the first dose (baseline) and at least once in the course of the following six weeks after the first dose. fCP levels were measured with the BUHLMANN fCAL ELISA kit (Buhlmann Diagnostics Corp, Amherst, NH) according to the manufacturer’s instructions. fCP is found in neutrophils and monocytes and its elevation in the feces is associated with the degree of endoscopic and histological inflammatory activity in IBD [11]. A relevant fCP increase was defined as a value above 250 µg/g and at least 50% greater than the baseline. If the baseline fCP level was above 250 µg/g, a relevant increase was described by a surge of the level of at least 50% only. The choice of this cut-off value was based on the data from previous studies demonstrating that fCP < 250 µg/g is associated with the absence of intestinal ulceration [12] and therefore reflective of a state of endoscopic improvement/remission. Furthermore, inter-sample variability of fCP levels from 2 specimens of the same stool sample is <15% in our laboratory (data not shown).

Analogously to the fCP measurement, serum C-reactive protein (CRP) was measured at the baseline and at least once in the six weeks following the first immunization. As this parameter was considered solely as an exploratory outcome reflecting systemic inflammation [13], no definition of a significant increase was set. The cut-off value at our center is ≤0.5 mg/dL. CRP levels were measured with an immunoturbidimetric assay of the Cobas 8000 system according to the manufacturer’s instruction (Roche, Mannheim, Germany).

IBD-related adverse events of the vaccination were recorded (new onset or worsening of increased stool frequency, development of abdominal pain, or rectal bleeding) until six weeks after the first dose. Four weeks after the second dose, levels of anti-SARS-CoV-2 antibodies against the spike protein in serum were measured with the Anti-SARS-CoV-2-QuantiVac-ELISA (IgG) (EUROIMMUN, Luebeck, Germany). The ELISA tests were performed at the Institute for Hygiene and Applied Immunology, Center for Pathophysiology, Infectiology and Immunology, according to the manufacturer’s instructions.

Prior to the first vaccination and one week after the second dose, peripheral blood mononuclear cells (PBMC) were isolated using Ficoll™ (LSM 1077, PAA, Pasching, Austria) according to the standard protocol [14]. PBMC were stimulated either with peptide pools of the S1 domain of the viral spike protein or the nucleocapsid protein (PepTivator^®^, Miltenyi biotech, Bergisch Gladbach, Germany) for 24 h. Incubation with PHA served as a positive control and 1xDPBS (GibcoTM, Thermo Fischer, Vienna, Austria) as a negative control. The concentration of interleukin (IL)-2, IL-5, IL-10, granulocyte-macrophage colony-stimulating factor (GM-CSF), and interferon-γ (IFNγ) were analyzed with Luminex^®^ 100/200 system (LuminexCorp, Austin, TX, USA).

Continuous parameters are given as medians with ranges or interquartile ranges. For categorical variables, percentages are reported. Statistical analyses were performed in GraphPad Prism Version 9.2.0. (San Diego, CA, USA), Stata 17 (Stata Corp. Colle Station, TX, USA) and included a paired *t*-test to test the H0: delta fCP = 0, the Fisher’s exact test to test the association between the categorized variables, and exact logistic regression to estimate the effect of the fCP increase on symptoms, as well as the effect of therapy on an increase in fCP after vaccination, and linear regression to estimate the effect of fCP on the antibody titer. We considered a two-sided *p*-value ≤ 0.05 as statistically significant.

The study was approved by the Ethics Committee of the Medical University Vienna, Austria (approval number 1073/2021) and conducted in accordance with Good Clinical Practice Guidelines. All research participants provided written informed consent.

## 3. Results

### 3.1. Study Population

In total, 42 patients were enrolled (45% females), including 24 with a diagnosis of Crohn’s disease (57.1%), 17 with ulcerative colitis (40.5%), and one with IBD unclassified (2.4%) (Table 1). One patient had had a previous COVID-19 infection. Most patients received anti-TNF agents (infliximab 6 (14.3%); adalimumab 11 (26.2%), golimumab 3 (7.1%)) or other biologics (ustekinumab 6 (14.3%), vedolizumab 5 (11.9%)). Six subjects (14.3%) were on immunosuppressants (azathioprine, methotrexate). Patients could take more than one medication at once. The ongoing therapy with biologicals or immunosuppressants was not a contraindication to the vaccination and was in line with national vaccination guidelines [15].

### 3.2. The Course of Fecal Calprotectin and Association with IBD-Related Adverse Events

At baseline, 21 subjects (21/42, 50%) were in clinical remission, defined as stool frequency ≤ 3/day and a daily abdominal pain score ≤ 1 for Crohn’s disease and stool frequency ≤ 1, and a rectal bleeding score < 1 for ulcerative colitis [16]. IBD-related adverse events (AE) were reported in four (9.5%) patients (two subjects each with ulcerative colitis and Crohn’s disease), two patients each after the first and second dose. All of them were in clinical remission at baseline. Apart from a single individual with IBD-related AEs after the first vaccine, all others displayed clinically relevant increases in fCP. Symptoms resolved in all cases without treatment escalation, within seven days in three patients and in one after four weeks. An alleviation of clinical symptoms was observed in one patient after administration of the first dose.

The median baseline fCP in all patients was 102.0 µg/g (range from 0–4727 µg/g; median time of sampling—0 days before the first dose, range 0–83 days) with 24 of them (57.1%) having fCP values < 250 µg/g. There was no difference (*p* = 0.13, df = 39) between baseline fCP levels and highest values (median value 217 µg/g, range 0–4850 µg/g) ever measured within the following six weeks after the first vaccine dose.

In 24 patients (57.1%) a post-baseline fCP after the first and prior to the second dose was available (median time of sampling from the first dose—26.5 days, range 3–28 days). In those patients, median fCP levels (median baseline fCP 69 µg/g; range 0–4727 µg/g) remained stable after the first vaccine dose (median fCP 54.5 µg/g; range 0–2324 µg/g; *p* = 0.61, df = 23) (Figure 1). However, in five (5/24, 20.8%) patients the pre-defined criterion of an increase in fCP levels was fulfilled (median baseline fCP 332 µg/g, range 41–993 µg/g; median post first dose fCP 584 µg/g, range 435–1699 µg/g).

In 29 (69.0%) patients, a post-baseline fCP after the second dose was available (median time from the second dose—four days, range 1–11 days). The median fCP level increased significantly from 80 µg/g at baseline (range 0–3344 µg/g) to 174 µg/g (range 0–4850 µg/g; *p* = 0.0084, df = 28) (Figure 1). Eight patients (8/29, 27.6%) achieved the pre-specified criterion of an fCP increase (median baseline fCP 277.5 µg/g, range 49–1522 µg/g; median post second dose fCP—1744/µg/g; range 259–3179 µg/g).

Concomitant treatment with vedolizumab was associated with an increase in fCP changes after the second vaccine dose (OR 12.4, 95% CI [1.6; 120.2], *p* = 0.0171, df = 40). In a univariate analysis, none of the other parameters (age, diagnosis, location, smoking, baseline CRP, baseline albumin, sex) was associated with an fCP increase.

For five patients who experienced a pre-specified fCP change either after the first or second dose, follow-up stool samples were collected within two weeks of the second inoculation. The fCP values measured showed a trend towards normalization—a decline of 43.0 to 97.7% compared to the highest fCP observed post-vaccination.

The increases in fCP levels were not associated with IBD-related AEs (for the changes after the first dose—*p* = 0.06, df = 41; for the changes after the second dose—*p* = 0.53, df = 41). Furthermore, there was no association between the IBD-related AEs and any biological or immunosuppressive medication (*p* > 0.05 for all medication types).

### 3.3. Exploratory Outcomes

At baseline, serum CRP from 38 (90.5%) patients could be obtained (median CRP—0.15 mg/dL; range 0.03–1.54 mg/dL; median time of sampling—0 days before the first dose, range 0–56 days). In the period between the first and second immunization, blood samples from 17 (40.5%) patients were collected (median CRP—0.19 mg/dL, range 0.03–0.77 mg/dL; median time of sampling—28 days after the 1st dose, range 3–28 days). There was no statistically significant difference between these two collection time points (*p* = 0.1837, df = 16) (Figure 2a). Within two weeks after the second mRNA-1273 dose, CRP values from 10 (23.8%) patients that had not had any blood test since the baseline could be obtained (median CRP—0.15 mg/dL; range 0.1–1.21 mg/dL; median time of sampling—1 day after the 2nd dose, range 1–3 days). Again, this change was not significant compared with the levels prior to the first inoculation (*p* = 0.1447, df = 9). However, 7 out of these 10 patients experienced a numeric increase in CRP (range 7.7–3266.7%) (Figure 2b). This resembles a similar behavior of the fCP course after the second dose where a numeric increase was observed in most participants (22/29, 75.9%) but only 8 patients fulfilled a pre-defined criterion.

### 3.4. SARS-CoV-2 Antibody Testing

Sera from 38 patients (90.5%) were obtained for the measurement of antibodies against the viral spike (S) protein four weeks after the second vaccine dose. All subjects developed antibodies with a median titer of 2223 BAU/mL (range 113.5—7008 BAU/mL). Antibody titers did not correlate with the fCP increase after the second vaccination (*p* = 0.844, df = 37) (Figure 3). However, there was a weak association between antibody titers and an fCP increase after the first dose (*p* = 0.04, df = 37).

### 3.5. Cellular Response after COVID-19 Vaccination

Cellular response was assessed by the measurements of cytokines in the supernatants from the stimulated PBMC prior to the first dose and one week after the second dose. Both samples could be obtained from 15 (35.7%) patients. All of them experienced an increase in IL-2, IL-5, IL-10, GM-CSF, and IFNγ levels (*p* < 0.05) (Figure 4). The cytokine concentrations were not associated with fCP increase after the second dose. However, only 2 patients in this group experienced a significant fCP rise.

## 4. Discussion

Seeking objective evidence of an increase in mucosal inflammatory burden after vaccination for SARS-CoV-2 in patients with inflammatory bowel disease, we assessed intestinal tolerability and fCP levels during the course of immunization with the mRNA-1273 Moderna vaccine. To the best of our knowledge, we describe for the first time a significant increase in fCP levels during standard two-dose vaccination with an mRNA vaccine in IBD patients. Our finding is of clinical relevance, as 3 out of 4 patients who presented with symptomatic intestinal deterioration also experienced increasing fCP levels.

The pivotal trial on the mRNA-1273 vaccine reported an 8.2% rate of adverse events related to the gastrointestinal tract (defined there as nausea and vomiting) in baseline SARS-CoV-2 negative patients [8]. Adverse events that could be attributed to a direct intestinal reactivity were not described in the pivotal trial. The rate of gastrointestinal adverse events in 9.5% of individuals with IBD (4.8% after the first dose, 4.8% after the second dose) in our study is similar to the rate reported in another study [9]. From our results, we were not able to significantly associate post-vaccine deterioration of intestinal patient-reported symptoms with increases in fCP levels, which were particularly pronounced after the second vaccine dose, mostly due to sampling size limitation. However, whereas 31% of patients achieved our pre-defined increase in fCP levels after any dose, only 9.5% flared, which is reflective of a much higher potential in mucosal reactivity as compared to the clinical patient perception. Consequently, patients and physicians need to be aware of potential changes in the clinical presentation of patients with IBD and their inflammatory biomarkers after vaccination. Although we were not able to compare fCP levels after vaccination in a healthy population, the robustness of our results can be confirmed by inter-sample variability from the same bowel movement in fCP measurements—the coefficient of variability at our center is 11.9% (unpublished data), whereas the threshold chosen to identify a significant fCP change is 50%. Nonetheless, as clinical deteriorations were only transient, without prompting treatment escalation, and fCP levels tended to drop following the previous rise in the few patients with further follow-up, we presumed that the enhanced immune response by the intestinal mucosa is mostly confined to a period of few days after vaccination, in particular the second dose. Corresponding results after a third vaccine would be of further interest.

Whether the mechanism triggering post-SARS-CoV-2 vaccination associated increases in fCP levels relates to unspecific immune activation or the fact that SARS-CoV-2 interacts via the angiotensin-converting enzyme 2 (ACE2) as its entry receptor [17], which is abundantly expressed in the intestinal mucosa [18], deserves further scrutiny. ACE2 is a key regulator of dietary amino acid homeostasis and its deficiency results in epithelial damage, which increases the susceptibility to intestinal inflammation [19]. During infection with SARS-CoV-2 the viral spike protein also downregulates ACE 2, which may contribute to intestinal symptoms and inflammation in COVID-19 patients in whom elevated levels of fCP have been described [18,20,21,22]. Available mRNA vaccines encode the full SARS-CoV-2 spike ectodomain [23]. The intestinal expression of the mRNA vaccine-induced spike protein and its potential role as an inducer of mucosal ACE2 needs to be determined.

In concordance with previous studies [1], we observed anti-SARS-CoV-2 antibodies against the spike protein in all our patients. We observed a minor association between the titer of the anti-spike protein antibodies and increased fCP levels only after the first vaccination dose, but not the second, which makes at least the humoral axis of the anti-SARS-CoV-2 vaccine-induced immune response an unlikely mediator of our observed post-vaccination intestinal immune activation. Moreover, as there was no difference in antibody levels between subjects with an fCP rise at any time and those without relevant fCP changes, the extent of the evoked intestinal reaction does not seem to have an impact on a long-lasting response to the vaccination. Nor did the fCP rise correlate with the extent of cellular response assessed by the levels of growth factor IL-2 or Th1-specific, antiviral cytokine IFNγ [24]. Thus, factors that modulate the immunization efficacy as evaluated by antibody titer require further scrutiny.

Anti-cytokine biologics (infliximab, adalimumab, ustekinumab) are known to reduce the expression levels of ACE2 [18], which may contribute to the lack of association between the risk of SARS-CoV-2 infection and in general treatment with biologicals [3] in patients with IBD. However, Khan et al. have described an increased risk of infection in IBD patients treated with the anti-adhesion antibody vedolizumab [25]. Interestingly, among baseline patient and disease characteristics that we analyzed in univariate analysis, only treatment with vedolizumab was associated with an increased risk of vaccination-associated fCP-level increases. It was also vedolizumab that elicited severe systemic reaction to the second vaccination dose in a larger American cohort study [26]. However, due to our small sample size, this observation needs confirmatory studies.

Our study is not without limitations. Whether our findings can be extrapolated to SARS-CoV-2 vaccines other than mRNA-1273 remains to be determined. Furthermore, it needs to be explored whether the timing of stool sampling relative to both inoculations was most appropriate to measure the full spectrum of post-vaccination changes in fCP levels. Nonetheless, our finding of post-vaccination enhanced fCP levels is further corroborated by numerical in-creases in serum CRP concentrations after the second vaccination. Even though our findings may be confined by small size, they should spur further interest in the exploration of vaccine-induced intestinal inflammation in predisposed individuals.

## 5. Conclusions

In summary, immunization with the mRNA-1273 SARS-CoV-2 vaccine may lead to a significant increase in fCP levels in patients with inflammatory bowel disease. As a probable underlying mechanism, we suggest a transient mRNA vaccine-induced intestinal expression of the spike protein with consecutive down-regulation of ACE2. Further studies are needed to assess the potential, even though unlikely, long-term impact of our findings on the course of IBD. Nonetheless, our work also adds to the use of fCP as a biomarker to monitor the impact of vaccines on bowel inflammation.

## Figures and Tables

**Figure 1 vaccines-10-00759-f001:**
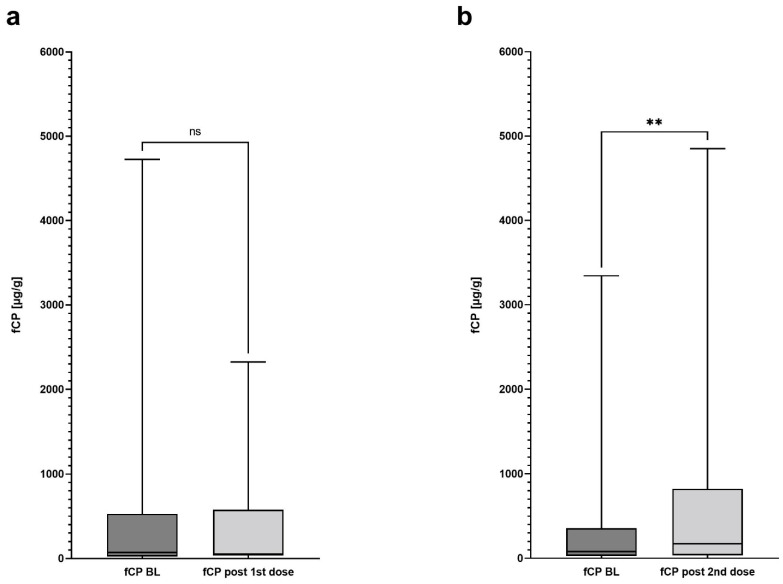
Difference in fecal calprotectin levels before and after mRNA-1273 vaccination. (**a**) The boxplots compare median values of baseline fecal calprotectin of 24 patients with the values after the first vaccination dose. No significant difference could be observed. (**b**) The boxplots compare median values of baseline fecal calprotectin of 29 patients with the values after the second vaccination dose. In both panels, (**a**,**b**), baseline is defined as a time point before the first dose. BL—baseline, fCP—fecal calprotectin, ns—not significant, **—*p* < 0.01.

**Figure 2 vaccines-10-00759-f002:**
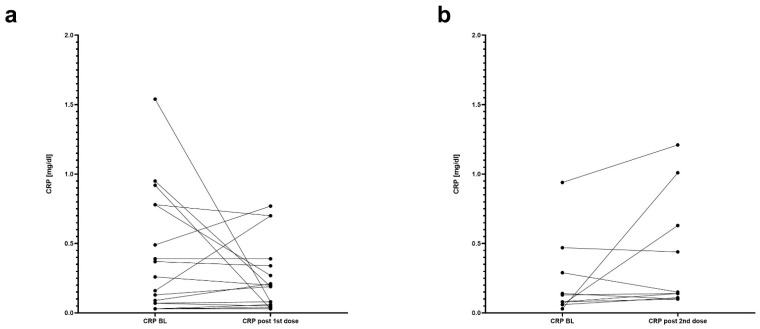
Changes in C-reactive protein levels before and after mRNA-1273 vaccination. (**a**) The panel compares values of baseline C-reactive protein of 17 patients with values after the first vaccination dose and prior to the second one. Each point represents another patient. (**b**) The panel compares values of baseline C-reactive protein of 10 patients with values after the second vaccination dose. Each point represents another patient. In both panels, (**a**,**b**), baseline is defined as a time before the first dose. BL—baseline, CRP—C-reactive protein.

**Figure 3 vaccines-10-00759-f003:**
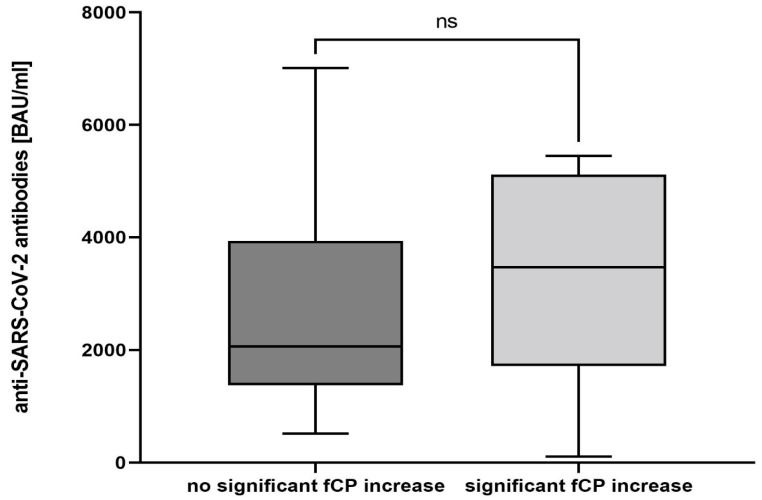
Difference in antibody against SARS-CoV-2 viral spike protein titers after mRNA-1273 vaccination. The boxplots compare median antibody levels four weeks after the second mRNA-1273 dose in patients that experienced a significant increase in fecal calprotectin at any time in the course of 6 weeks following the first vaccination dose (*n* = 11) with those who did not experience a relevant change in fecal calprotectin (*n* = 28). fCP—fecal calprotectin, ns—not significant.

**Figure 4 vaccines-10-00759-f004:**
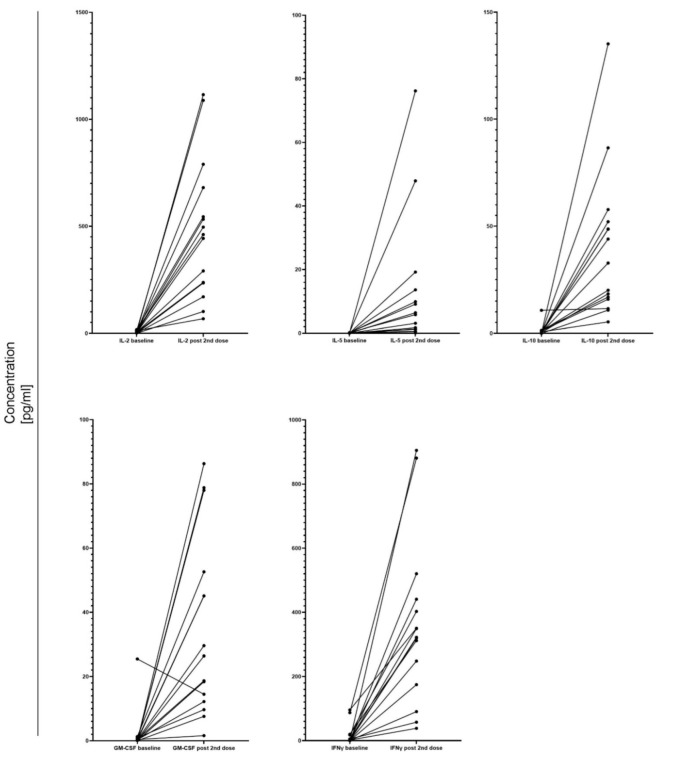
Changes in cytokine levels before and after mRNA-1273 vaccination. The figure compares median baseline cytokine levels of 15 patients with the values one week after the second mRNA-1273 dose. The bars represent interquartile ranges. IL-2 baseline: median 3.4, IQR 1.6–10.3 pg/mL; IL-2 post 2nd dose: median 461.5, IQR 67.8–680.7 pg/mL; IL-5 baseline: median 0.0, IQR 0.0–0.1 pg/mL; IL-5 post 2nd dose: median 5.8, IQR 1.2–13.6 pg/mL; IL-10 baseline: median 0.4, IQR 0.0–1.1 pg/mL; IL-10 post 2nd dose: median 32.8, IQR 15.9–52.1 pg/mL; GM-CSF baseline: median 0.3, IQR 0.0–0.5 pg/mL; GM-CSF post 2nd dose: median 26.4, IQR 12.2–51.3 pg/mL; IFNγ baseline: median 1.4, IQR 0.2–17.8 pg/mL; IFNγ post 2nd dose: median 322.0, IQR 174.8–440.0 pg/mL. Pairwise comparisons represent the results of paired t-tests. IL-2–interleukin-2, IL-5–interleukin-5, IL-10–interleukin-10, GM-CSF–granulocyte-macrophage colony-stimulating factor, IFNγ–interferon-γ, IQR–interquartile range.

**Table 1 vaccines-10-00759-t001:** Patient and disease characteristics.

Characteristic	All Patients (N = 42)
Age [years]—median (IQR)	41.5 (31.0–52.3)
Female sex—no. (%)	19 (45.0)
IBD type—no. (%):	
-Crohn’s disease	24 (57.1)
-Ulcerative colitis	17 (40.5)
-IBDU	1 (2.4)
Medication at the BL—no. (%):	
-Mesalazine	26 (61.9)
-Adalimumab	11 (26.2)
-Infliximab	6 (14.3)
-Ustekinumab	6 (14.0)
-Vedolizumab	5 (11.9)
-Azathioprine	4 (9.5)
-Corticosteroids	3 (7.1)
-Golimumab	3 (7.1)
-Methotrexate	1 (2.4)
-Probiotic Escherichia coli Strain Nissle 1917	1 (2.4)
Disease duration [years]—median (IQR):	
-Crohn’s disease	14.2 (6.5–20.3)
-Ulcerative colitis	8.5 (4.0–23.8)
Clinical remission at the BL—no. (%):	
-Crohn’s disease	16 (38.1)
-Ulcerative colitis	4 (9.5)
-IBDU	1 (2.4)
fCP [µg/g]—median (IQR):	
-baseline	102.0 (36.5–383.0)
-value after the 1st dose	54.5 (33.5–577.0)
-value after the 2nd dose	174.0 (37.5–823.5)
CRP [mg/dL]—median (IQR):	
-baseline	0.15 (0.03–0.49)
-value after the 1st dose	0.19 (0.03–0.37)
-value after the 2nd dose	0.15 (0.10–0.73)
IBD-related adverse events:	
-after the 1st dose	2 (4.8)
-after the 2nd dose	2 (4.8)

Clinical remission was defined as stool frequency ≤ 3/day and daily abdominal pain score ≤ 1 in Crohn’s disease and stool frequency ≤ 1 and rectal bleeding score = 0 in ulcerative colitis. Biomarker remission was defined as fecal calprotectin value < 50 µg/g. BL—baseline, fCP—fecal calprotectin. Patients could take more than one medication at once. For continuous variables, interquartile range is stated in the brackets. CRP—C-reactive protein, fCP—fecal calprotectin. IBD—inflammatory bowel disease. IBDU—indeterminate colitis, IQR—interquartile range.

## Data Availability

Data, analytic methods, and study material will be available on reasonable request and can be obtained from the corresponding author.

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
