# Peer review of "Course of Fecal Calprotectin after mRNA SARS-CoV-2 Vaccination in Patients with Inflammatory Bowel Diseases"

_vaccines, 2022, doi:10.3390/vaccines10050759_

Round 1
Reviewer 1 Report
The paper is interesting and extremely current. It is a field of great interest, because in real practice we often observe exacerbation of IBD after SARS-COV 2 vaccination.

Author Response
Dear Editor in Chief,
Thank you for reviewing our manuscript " COURSE OF FECAL CALPROTECTIN AFTER mRNA SARS-CoV-2 VACCINATION IN PATIENTS WITH INFLAMMATORY BOWEL DISEASES" and raising the opportunity to submit a revised version. We have provided detailed responses to reviewer’s comments below and highlighted the corresponding changes in the manuscript. Furthermore, we are submitting a clean copy as requested. We hope that you will now find our manuscript suitable for publication in Vaccines.
Reviewer 1
- What is the main question addressed by the research? The authors underline correlation in pathways of SARS-CoV-2 antigen expression and the pathogenesis of IBD because they show that immunization with the mRNA-1273 SARS-CoV-2 vaccine may lead to a significant increase in fCP levels in these patients
Authors’ reply: Many thanks for agreeing on the main research question addressed.
- Do you consider the topic original or relevant in the field, and if so, why? The topic is original and relevant because in literature there are no similar reports.
Authors’ reply: Many thanks for agreeing on the originality and relevance of our research topic.
- What does it add to the subject area compared with other published material? The work adds the use of a biomarker (fCP) as a mean to explain the impact of vaccine on bowel inflammation.
Authors’ reply: Many thanks for raising that point. Accordingly, we added the following sentence to the conclusion of our discussion: “Nonetheless, our work also adds to the use of fCP as a biomarker to monitor the impact of vaccines on bowel inflammation.”
- What specific improvements could the authors consider regarding the methodology?
Maybe they could increase the number of subjects in the study.
Authors’ reply: The reviewer points to the sample size of our study that we indicated as limitation in the discussion of the initial version of our manuscript. As a matter of fact, the presented results are derived from a sub-study on a larger program on the safety and immunogenicity of the mRNA-1273 SARS-CoV-2 vaccine in patients from various indications under immune-modulatory and -suppressive therapy at the Medical University of Vienna initiated early after the availability of that vaccine in Austria. The sub-study was triggered by a routine visit observation of an increased fCP after vaccination in a patient with IBD relatively late in the course of the main program which explains the limited sample size. Nonetheless, in line with the previous comments by reviewer 1 we do believe that our findings are original and relevant to the field and should trigger further studies on the impact of vaccination of intestinal inflammation in predisposed individuals. In order to address the reviewer’s comment further we added the following sentence to the discussion of the revised manuscript: “Even though our findings may be confined by small size, they should spur further interest in the exploration of vaccine-induced intestinal inflammation is predisposed individuals.”
- Are the conclusions consistent with the evidence and arguments presented and do they address the main question posed? Yes, they are.
Authors’ reply: We are grateful for the reviewer’s positive assessment.
- Are the references appropriate? Yes, they are.
Authors’ reply: We are grateful for the reviewer’s positive assessment.
- Please include any additional comments on the tables and figures. Tables and figure are very clear and explanatory
Authors’ reply: We are grateful for the reviewer’s positive assessment.
Reviewer 2 Report
The authors of the manuscript entitle " Course of fecal calprotectin after mRNA SARS-CoV-2 vaccination in patients with inflammatory bowel diseases" studied the effect of mRNA vaccine on inflammatory bowel diseases. They studied 42 patients with IBD and measured their fCP level before and after vaccination. Outcome of their study showed vaccination will cause a significant increase in fCP level in IBD patients.
The study design is proper and the authors measure numbers of parameters in their study. The results written nicely in details and easy to understand. Overall the study looks scientific and I have no comments to improve their manuscript.
Author Response
Dear Editor in Chief,
Thank you for reviewing our manuscript " COURSE OF FECAL CALPROTECTIN AFTER mRNA SARS-CoV-2 VACCINATION IN PATIENTS WITH INFLAMMATORY BOWEL DISEASES" and raising the opportunity to submit a revised version. We have provided detailed responses to the reviewer’s comments below and highlighted the corresponding changes in the manuscript. Furthermore, we are submitting a clean copy as requested. We hope that you will now find our manuscript suitable for publication in Vaccines.
The authors of the manuscript entitle " Course of fecal calprotectin after mRNA SARS-CoV-2 vaccination in patients with inflammatory bowel diseases" studied the effect of mRNA vaccine on inflammatory bowel diseases. They studied 42 patients with IBD and measured their fCP level before and after vaccination. Outcome of their study showed vaccination will cause a significant increase in fCP level in IBD patients.
The study design is proper and the authors measure numbers of parameters in their study. The results written nicely in details and easy to understand. Overall the study looks scientific and I have no comments to improve their manuscript.
Authors’ reply: Thank you for the favourable review.
Reviewer 3 Report
Thank you very much for allowing me to review the article entitled “Course of fecal calprotectin after mRNA SARS-CoV-2 vaccination in patients with inflammatory bowel diseases” (Vaccines-1681577).
This manuscript is submitted for publication within the “COVID-19 Vaccines and Vaccination” section.
The aim of this work is to assess the mucosal inflammatory activity in patients with IBD before and after immunization with mRNA-1273 (Moderna) vaccine by measurement of fecal calprotectin (fCP).
Comments: The introduction is very brief, it should be considered in a broader way, given the importance of the subject. Especially pointing to the biological plausibility sought.
This is a prospective study of patients with IBD (Inflammatory bowel diseases). In the summary it is identified that there are 42 patients who participate in the study but in material and methods it should be indicated how many patients are treated at the Medical University of Vienna, what was the participation rate and if those who accepted presented any special characteristics. participate that could be considered in the interpretation of the results. Have you carried out a calculation of the sample size? If so, please incorporate it into the manuscript.
It is therefore a series of cases. Have you considered using a reference group that does not present IBD to assess whether it is associated with the baseline situation (IBD) or is it an evolution in the entire population of the vaccination process?
The study is approved by the ethics committee.
It is not well understood why a patient who previously had COVID-19 is being kept. Please explain it.
In Table 1, the frequency presented does not add up to 42 patients. Please clarify the tables.
The discussion does not consider the limitations of the study and the need for future studies to confirm the trends presented. Especially since it is a series of cases.
The conclusions should be more limited to the results obtained in the context that it is a series of cases.
Author Response
Dear Editor in Chief,
Thank you for reviewing our manuscript " COURSE OF FECAL CALPROTECTIN AFTER mRNA SARS-CoV-2 VACCINATION IN PATIENTS WITH INFLAMMATORY BOWEL DISEASES" and raising the opportunity to submit a revised version. We have provided detailed responses to the reviewer’s comments below and highlighted the corresponding changes in the manuscript. Furthermore, we are submitting a clean copy as requested. We hope that you will now find our manuscript suitable for publication in Vaccines.
Reviewer 3
Thank you very much for allowing me to review the article entitled “Course of fecal calprotectin after mRNA SARS-CoV-2 vaccination in patients with inflammatory bowel diseases” (Vaccines-1681577).
This manuscript is submitted for publication within the “COVID-19 Vaccines and Vaccination” section.
The aim of this work is to assess the mucosal inflammatory activity in patients with IBD before and after immunization with mRNA-1273 (Moderna) vaccine by measurement of fecal calprotectin (fCP).
Comments: The introduction is very brief, it should be considered in a broader way, given the importance of the subject. Especially pointing to the biological plausibility sought.
Authors’ reply: We are grateful for the reviewer’s assessment on the importance of the subject of our study. Accordingly, we elaborated further on its biologic plausibility in the introduction of the revised manuscript as follows:
“So far, for patients with inflammatory bowel diseases (IBD) an increased risk of infection with SARS-CoV-2 has not been demonstrated [1], though some medications, in particular corticosteroids, have been associated with an enhanced risk of severe COVID-19 outcome [2,3]. As immunomodulatory medication is known for putting the patients at the increased risk of serious infections [4], prevention of contagion plays a crucial role in their daily life. Despite the availability of SARS-CoV-2 vaccines and early evidence of their efficacy in patients with IBD, concerns about the safety related to immune-mediated disease enhancement need to be scientifically addressed [5,6]. Adverse events - including gastrointestinal symptoms - were reported from the clinical trials with mRNA vaccines [7,8] as well as in a series of IBD patients [9]. Increased stool frequency, abdominal pain and rectal bleeding are the main symptoms of patients with IBD [10], but particularly the former two are less specific and could have various causes. Therefore, it would be relevant to know as to whether vaccination-induced gastrointestinal symptoms in patients with IBD are derived from an increase of the intestinal inflammation and to shed light on their further course. Corresponding insights might inform the decision on timing of vaccinations relative to the intestinal inflammatory burden.
During a routine follow-up in a patient with ulcerative colitis, we observed an increase in fecal calprotectin (fCP), the most sensitive and widely used biomarker for monitoring of intestinal inflammation in patients with IBD, immediately after vaccination with the mRNA-1273 SARS-CoV-2 vaccine. Over the recent years fCP has supplanted ileo-colonoscopy, which as invasive procedure is inadequate for rapid and repetitive monitoring in patients with IBD. We initiated the present study on patients with IBD, as a sub-study of a larger trial on the safety and immunogenicity of mRNA-1273 SARS-CoV-2 vaccine in immunocompromised patients, to explore changes in the intestinal inflammatory activity as measured by fCP during the course of the two-dose initiation SARS-CoV-2 vaccination.”
This is a prospective study of patients with IBD (Inflammatory bowel diseases). In the summary it is identified that there are 42 patients who participate in the study but in material and methods it should be indicated how many patients are treated at the Medical University of Vienna, what was the participation rate and if those who accepted presented any special characteristics. participate that could be considered in the interpretation of the results. Have you carried out a calculation of the sample size? If so, please incorporate it into the manuscript.
Authors’ reply: In line with our response to reviewer 1, the presented results are derived from a sub-study on a larger program on the safety and immunogenicity of the mRNA-1273 SARS-CoV-2 vaccine in patients from various indications under immune-modulatory and -suppressive therapy at the Medical University of Vienna initiated early after the availability of that vaccine in Austria. We added that information to the Methods section of the revised manuscript as follows:
“Patients with IBD treated at the Medical University of Vienna and participated in a study on the safety and immunogenicity of mRNA-1273 vaccine in immunocompromised patients were enrolled in our sub-study. Due to the exploratory character of the study a formal sample size estimate was not performed.”
In total about 200 patients with IBD consented to participate in the main study. Patients were recruited with a focus on those who were under treatment with anti-TNF agents or the anti-integrin biologic, vedolizumab. As this sub-study was triggered by a routine visit observation of an increased fCP after vaccination in a patient with IBD relatively late in the course of the main program, we succeeded to still recruit 42 patients. Nonetheless, in line with the previous comments by reviewer 1 we do believe that our findings are original and relevant to the field and should trigger further studies on the impact of vaccination of intestinal inflammation in predisposed, but also healthy individuals.
It is therefore a series of cases. Have you considered using a reference group that does not present IBD to assess whether it is associated with the baseline situation (IBD) or is it an evolution in the entire population of the vaccination process?
Authors’ reply: As the present study was an amendment to the main study which included only immunocompromised patients, the inclusion of non-IBD controls was not possible, even though initially considered as a potentially valuable addition to the program. We added that thought to the discussion of our revised manuscript: “Even though our findings may be confined by small size, they should spur further interest in the exploration of vaccine-induced intestinal inflammation in pre-disposed, but also healthy individuals.”
The study is approved by the ethics committee.
It is not well understood why a patient who previously had COVID-19 is being kept. Please explain it.
Authors’ reply: We agree with the reviewer that this appears counterintuitively with the current knowledge on the necessity of vaccinating also individuals who contracted SARS-CoV-2 infection before. However, as described above, the presented results are derived from a sub-study on a larger program on the safety and immunogenicity of the mRNA-1273 SARS-CoV-2 vaccine in patients from various indications under immune-modulatory and -suppressive therapy at the Medical University of Vienna initiated early after the availability of that vaccine in Austria. In order to interrogate the immunogenicity of the vaccine in an at that time unexplored patient population the members of the steering committee of the main study decided to recruit only individuals without prior infection with SARS-CoV-2.
In Table 1, the frequency presented does not add up to 42 patients. Please clarify the tables.
Authors’ reply: As patients could be treated with more than one drug concomitantly, medications indeed don’t add up to the number of the entire patient population of 42. The percentages of patients in clinical remission at baseline refer to that number, which was added to the footnote of the table.
The discussion does not consider the limitations of the study and the need for future studies to confirm the trends presented. Especially since it is a series of cases.
Authors’ reply: According to the comment we expanded the paragraph on limitations in the way:
“Our study is not without limitations. Whether our findings can be extrapolated to SARS-CoV-2 vaccines other than mRNA-1273 remains to be determined. Furthermore, it needs to be explored whether the timing of stool sampling relative to both inoculations was most appropriate to measure the full spectrum of post-vaccination changes in fCP levels. Nonetheless, our finding of post-vaccination enhanced fCP levels is further corroborated by numerical in-creases in serum CRP concentrations after the second vaccination. Even though our findings may be confined by small size, they should spur further interest in the exploration of vaccine-induced intestinal inflammation in predisposed, but also healthy individuals.”
The conclusions should be more limited to the results obtained in the context that it is a series of cases.
Authors’ reply: The paragraph on conclusions was revised accordingly:
“In summary, immunization with the mRNA-1273 SARS-CoV-2 vaccine may lead to a significant increase in fCP levels in patients with inflammatory bowel disease. The worsening of IBD symptoms observed post-vaccination was accompanied by an elevation of fCP in 75% cases. As a probable underlying mechanism, we suggest a transient mRNA vaccine-induced intestinal expression of the spike protein with consecutive down-regulation of ACE2. Further studies are needed to assess the potential long-term impact of our findings on the course of IBD. Nonetheless, our work also adds to the use of fCP as a biomarker to monitor the impact of vaccines on bowel inflammation.”